# Mechanistic Insights into Melatonin’s Antiarrhythmic Effects in Acute Ischemia-Reperfusion-Injured Rabbit Hearts Undergoing Therapeutic Hypothermia

**DOI:** 10.3390/ijms26020615

**Published:** 2025-01-13

**Authors:** Hui-Ling Lee, Po-Cheng Chang, Hung-Ta Wo, Shih-Chun Chou, Chung-Chuan Chou

**Affiliations:** 1Department of Anesthesia, Chang Gung Memorial Hospital, Taipei Branch, Taipei 10507, Taiwan; ring3412@yahoo.com.tw; 2Division of Cardiology, Department of Internal Medicine, Chang Gung Memorial Hospital, Taoyuan Branch, Taoyuan 33304, Taiwan; pccbrian@gmail.com (P.-C.C.); hungtawo@gmail.com (H.-T.W.); kobechou2408@gmail.com (S.-C.C.); 3School of Medicine, Chang Gung University College of Medicine, Taoyuan 33302, Taiwan

**Keywords:** electrophysiology, ischemia-reperfusion injury, melatonin, therapeutic hypothermia, ventricular fibrillation

## Abstract

The electrophysiological mechanisms underlying melatonin’s actions and the electrophysiological consequences of superimposed therapeutic hypothermia (TH) in preventing cardiac ischemia-reperfusion (IR) injury-induced arrhythmias remain largely unknown. This study aimed to unveil these issues using acute IR-injured hearts. Rabbits were divided into heart failure (HF), HF+melatonin, control, and control+melatonin groups. HF was induced by rapid right ventricular pacing. Melatonin was administered orally (10 mg/kg/day) for four weeks, and IR was created by 60-min coronary artery ligation and 30-min reperfusion. The hearts were then excised and Langendorff-perfused for optical mapping studies at normothermia, followed by TH. Melatonin significantly reduced ventricular fibrillation (VF) maintenance. In failing hearts, melatonin reduced the spatially discordant alternans (SDA) inducibility mainly by modulating intracellular Ca^2+^ dynamics via upregulation of sarcoplasmic reticulum Ca^2+^-ATPase (SERCA2a) and calsequestrin 2 and attenuating the downregulation of phosphorylated phospholamban protein expression. In control hearts, melatonin improved conduction slowing and reduced dispersion of action potential duration (APD_dispersion_) by upregulating phosphorylated connexin 43, attenuating the downregulation of SERCA2a and phosphorylated phospholamban and attenuating the upregulation of phosphorylated Ca^2+^/calmodulin-dependent protein kinase II. TH significantly retarded intracellular Ca^2+^ decay slowed conduction, and increased APD_dispersion_, thereby facilitating SDA induction, which counteracted the beneficial effects of melatonin in reducing VF maintenance.

## 1. Introduction

The mechanism of cardiac ischemia-reperfusion (IR) injury-induced intracellular Ca^2+^ (Ca_i_) overload is multifactorial, including increase of intracellular Na^+^ concentration via the Na^+^/H^+^ exchanger and dysregulation of L-type Ca^2+^ channels, sodium-calcium exchanger (NCX), sarcoplasmic reticulum Ca^2+^-ATPase (SERCA2a), ryanodine receptor 2 (RyR2), and mitochondria [1]. The elevated Ca_i_ leads to Ca^2+^-dependent cellular apoptosis and contributes to focal and reentrant arrhythmias by promoting automaticity, triggered activity, and dispersion of excitability and refractoriness, respectively [2,3,4,5]. Furthermore, IR injury-induced downregulation of voltage-gated sodium current (*I*_Na_) [6] and gap junction uncoupling [7] slow conduction and increased dispersion of repolarization. These electrophysiological changes observed after reperfusion may lead to life-threatening ventricular tachyarrhythmias [8]. Melatonin, a circadian molecule with marked antioxidant properties, has been shown to preserve the microstructure of cardiomyocytes and reduce myocardial IR injury [9]. The anti-oxidative effects of melatonin might play a role in protecting against IR-induced arrhythmias. Tan et al. [10] were the first to demonstrate the beneficial effects of melatonin in isolated rat hearts subjected to IR-induced VA. Lee et al. [11] revealed that the antiarrhythmic effect of melatonin may be mediated by its antioxidant activity and neutrophil inhibition capacity in IR-injured rat hearts in vivo. Using a different animal model, Bertuglia et al. [12] reported that pretreatment of melatonin for 3 weeks prevented VA by reducing oxidative and nitrative stress in IR-injured cardiomyopathic hamster hearts. However, Sedova et al. [13] showed that the melatonin-related antiarrhythmic effect on ventricular activation was independent of its anti-oxidative properties in IR-injured rat hearts. Durkina et al. also reported that the antiarrhythmic effects of melatonin via the receptor-dependent enhancement of impulse conduction and maintenance of resting membrane potential are independent of its antioxidant properties [14]. Moreover, melatonin was reported to preserve SERCA2a expression and increase SERCA2a activity to reduce IR-induced Ca_i_ overload [15], but Durkina et al. [16] reported that pretreatment of melatonin for 3 weeks did not modify the extrasystolic burden in IR-injured rat hearts. Although the antiarrhythmic effects of melatonin in IR injury have been confirmed in several studies, the mechanisms of its antiarrhythmic effects and molecular targets are still not fully elucidated.

Therapeutic hypothermia (TH) ameliorates oxidative injury and interrupts the early stages of the apoptotic pathway during IR injury [17]. Combined treatment with melatonin and TH was reported to improve long-term neurodevelopment in asphyctic neonates [18]. Melatonin may act synergistically with TH to reduce cardiac IR injury. However, hypothermia-induced cell-to-cell uncoupling enhances action potential duration dispersion (APD_dispersion_), contributes to conduction slowing, and promotes the onset of spatially discordant alternans (SDA) [19]. Further, hypothermia retards SERCA2a-mediated Ca^2+^ uptake to facilitate Ca_i_ alternans induction [20]. It is unknown whether TH would counteract melatonin’s antiarrhythmic effects in acute IR-injured hearts, particularly in heart failure (HF). In this study, we performed optical mapping to investigate the antiarrhythmic mechanism of melatonin in isolated failing or controlled IR-injured rabbit hearts at normothermia and TH. Western blotting was performed to identify the molecular targets of melatonin’s antiarrhythmic actions in these models.

## 2. Results

After a 4-week pacing period, the LV ejection fraction decreased from 72 ± 3% to 45 ± 7% (*p* < 0.001) in the HF group and from 72 ± 3% to 44 ± 9% (*p* < 0.001) in the HF+melatonin group. The baseline LV ejection fraction was 73 ± 4% in the control group and 73 ± 2% in the control+melatonin group (*p* = NS). The heart weight was significantly heavier in HF (19.13 ± 2.66 g) and HF+melatonin (19.31 ± 2.12 g) than control (13.97 ± 2.34 g) and control+melatonin (13.83 ± 1.29 g) groups, respectively (*p* < 0.001 for both comparisons). The heart weight/body weight ratio was also significantly higher in HF (0.53 ± 0.08%) and HF+melatonin (0.55 ± 0.07%) than control (0.42 ± 0.07%) and control+melatonin (0.39 ± 0.05%), respectively (*p* < 0.001 for both comparisons).

### 2.1. Effects of Melatonin Pretreatment and TH on Electrophysiological Properties

#### 2.1.1. Ca_i_ Decay

The electrophysiological effects of melatonin and TH are summarized in Table 1. Ca_i_ decay time (τ value) was significantly longer in the IR zone than the non-IR zone at baseline, and TH significantly prolonged τ values in all four groups. In failing hearts, melatonin had no significant effect on Ca_i_ decay in the non-IR zone but fastened Ca_i_ decay in the IR zone at baseline (τ value 51.7 ± 3.5 ms vs. 59.4 ± 6.4 ms, *p* = 0.004) and TH (τ value 103.3 ± 11.3 ms vs. 118.8 ± 13.8 ms, *p* = 0.013, Figure 1A). Notably, there was no significant difference in τ value between the non-IR and IR zones at TH in the HF+melatonin group (*p* = 0.855). Figure 1B shows the representative Ca_i_ traces in failing hearts. In both hearts, τ value in the IR zone was longer than that in the non-IR zone and was prolonged by TH. Melatonin treatment resulted in smaller τ values in the IR zone at baseline (48.5 ms vs. 61.9 ms) and TH (105.4 ms vs. 150.7 ms) and a much smaller difference of τ value between the non-IR and IR zones at TH (0.2 ms vs. 36.5 ms). In control hearts, melatonin had no significant effect on Ca_i_ decay in the non-IR and IR zones. As shown in Figure 1C, there was no significant difference in τ values in the IR zone between the control and control+melatonin groups at baseline (50.0 ± 6.9 ms vs. 50.1 ± 5.2 ms, *p* = 0.965) and TH (103.3 ± 18.2 ms vs. 98.3 ± 15.3 ms, *p* = 0.515). Figure 1D shows the representative Ca_i_ traces in control hearts. Melatonin treatment resulted in slightly shorter τ values in the IR zone at baseline (51.0 ms vs. 54.7 ms) and TH (88.5 ms vs. 94.4 ms), and a smaller difference of τ value between the non-IR and IR zones at TH (1.5 ms vs. 7.0 ms).

#### 2.1.2. CV

CV_IR_ was significantly slower than CV_non-IR_, and TH significantly slowed CV in all four groups. In failing hearts (Figure 2A), melatonin had no significant effect on CV at baseline and TH. Figure 2B shows the representative isochronal maps at PCL = 300 ms in failing hearts. CV_IR_ was significantly slower than CV_non-IR_, and TH slowed both CV_non-IR_ and CV_IR_ in both groups. Melatonin slightly increased CV_non-IR_ and CV_IR_ at baseline (by 5 and 8 cm/s, respectively) and TH (by 3 and 8 cm/s, respectively). In control hearts (Figure 2C), the control+melatonin group had a significantly faster CV_IR_ than the control group at baseline (78.0 ± 10.2 cm/s vs. 61.5 ± 15.3 ms, *p* = 0.011) and TH (41.8 ± 3.1 cm/s vs. 34.4 ± 8.9 ms, *p* = 0.023), and faster CV_non-IR_ at TH (53.6 ± 5.6 cm/s vs. 46.9 ± 7.8 ms, *p* = 0.038). As shown in Figure 2D, melatonin increased CV_non-IR_ and CV_IR_ at baseline (by 11 and 12 cm/s, respectively) and TH (by 9 and 4 cm/s, respectively).

#### 2.1.3. ERP and APD

As summarized in Table 1, melatonin had no significant effects on ERP in failing hearts at baseline (*p* = 1) and TH (*p* = 0.704) and in control hearts at baseline (*p* = 0.335) and TH (*p* = 0.777). TH prolonged ERP in all four groups. Figure 3 summarizes the effects of melatonin and TH on APD. TH prolonged APD_max_ and APD_min_ and increased APD_dispersion_ in all four groups. In failing hearts (Figure 3A), melatonin had no significant effects on APD_max_, APD_min_, and APD_dispersion_ at baseline, but significantly prolonged APD_min_ (199 ± 17 ms vs. 180 ± 20 ms, *p* = 0.040) and decreased APD_dispersion_ (53 ± 14 ms vs. 66 ± 13 ms, *p* = 0.045) at TH. Figure 3B shows representative APD maps at PCL = 300 ms. Melatonin reduced APD_dispersion_ from 24 to 20 ms at baseline and from 70 to 42 ms at TH. The APD_min_ was much longer in the melatonin-treated failing heart at TH (208 ms vs. 184 ms). In control hearts (Figure 3C), melatonin significantly prolonged APD_min_ (109 ± 10 ms vs. 99 ± 9 ms, *p* = 0.040) and thus decreased APD_dispersion_ (25 ± 9 ms vs. 32 ± 7 ms, *p* = 0.043) at baseline, but had no significant effects on APD_max_, APD_min_, and APD_dispersion_ at TH. Representative APD maps (Figure 3D) showed that melatonin reduced APD_dispersion_ from 36 ms to 16 ms at baseline and from 48 ms to 36 ms at TH.

#### 2.1.4. SCA and SDA

As summarized in Table 1, SCA was inducible in all hearts at baseline and TH. Melatonin had no significant effect on SCA induction, while TH lowered the SCA threshold of SCA in all four groups. On the other hand, melatonin significantly suppressed SDA induction in failing hearts at baseline. As summarized in Figure 4A, SDA was induced in 8 of 10 and 2 of 10 hearts in the HF and HF+melatonin groups, respectively (*p* = 0.023), and in 9 of 10 and 5 of 10 hearts in the control and control+melatonin groups, respectively (*p* = 0.141). However, SDA was induced in all rabbits at TH; that is, melatonin was unable to suppress SDA induction at TH. Figure 4B shows representative V_m_, and Ca_i_ alternans maps in failing hearts. SDA was induced at PCLs of 110 ms at baseline and 260 ms at TH in a failing heart without melatonin pretreatment (upper panels). Melatonin suppressed SDA induction at baseline but not at TH. As shown in the bottom panels, SDA was not inducible even at PCL = 100 ms at baseline but was inducible at PCL = 300 ms at TH in a failing heart with melatonin pretreatment.

### 2.2. Effects of Melatonin and TH on VF Maintenance in Isolated Rabbit Hearts with Acute IR Injury

Figure 5 summarizes the effects of melatonin and TH on VF maintenance. In failing hearts (Panel A), melatonin reduced VF maintenance significantly from score 2.0 ± 0 to 0.4 ± 0.8 (*p* < 0.001) at baseline, but not at TH (1.4 ± 0.9 vs. 1.1 ± 0.8, *p* = 0.476); in control hearts (Panel B), melatonin reduced VF maintenance significantly from score 1.3 ± 0.9 to 0.2 ± 0.6 (*p* = 0.007) at baseline, but not at TH (0.5 ± 0.7 vs. 0.5 ± 0.8, *p* = 1). This implied that TH counteracted the beneficial effects of melatonin on VF suppression in both failing and control hearts. In the absence of melatonin, TH had no significant effects on VF maintenance (score 2.0 ± 0 vs. 1.4 ± 0.9, *p* = 0.081) in failing hearts but reduced VF maintenance from score 1.3 ± 0.9 to 0.5 ± 0.7 (*p* = 0.037) in control hearts.

Figure 6 shows a representative example of VF induction by extra stimulus pacing from the same heart, as shown in Figure 4B (HF group). Panel A shows IR model creation, and the mapping field was shown in Panel B. Panels C and D are pseudo-electrocardiograms and V_m_ tracings recorded at sites “a” (on the nodal line) and “b” (at the IR zone) during VF induction. As shown in panel E, extra stimulus pacing-induced a reentrant wavefront (beat S_5_), followed by epicardial breakthroughs from the upper border (beat “1”) and IR zone (beat “2”), and then reentrant wavefronts surrounding site “a” (beats “3” and “4”), thereby perpetuating VF. The corresponding V_m_ tracing at site “a” showed fragmented potentials during VF initiation.

Melatonin suppressed SDA induction, thereby reducing VF inducibility. Figure 7 shows a representative example from the same heart as shown in Figure 4B (HF+melatonin group). Panel A shows IR model creation. Panel B shows the mapping field and APD alternans map at PCL = 100 ms, in which no SDA was induced. VF was not inducible by extra stimulus pacing (Panels C and D). Panel E shows isochronal maps (upper panel) and phase maps (bottom panel) during extra stimulus pacing. Neither conduction block nor phase singularity was induced by extra stimulus pacing up to S_5_.

### 2.3. Effects of Melatonin Treatment on Protein Expression in Acute IR Injured Hearts

To elucidate the molecular targets of melatonin’s antiarrhythmic effects in IR-injured hearts, the levels of Ca^2+^-handling proteins, Na^+^ channel, and Cx43-associated proteins were measured and compared between the IR and non-IR zones. Figure 8A summarizes the results of Ca^2+^-handling protein expression. Melatonin upregulated SERCA2a expression (*p* = 0.035) and attenuated PLN-s (*p* = 0.283) and PLN-t (*p* = 0.061) downregulation in failing hearts and attenuated SERCA2a (*p* = 0.589) and PLN-s (*p* = 0.156) downregulation in control hearts. CASQ2 was downregulated in the IR zone of control hearts (*p* = 0.047). Melatonin failed to attenuate CASQ2 downregulation in control hearts (*p* = 0.028) but upregulated CASQ2 in failing hearts (*p* = 0.020). CaMKII-*p* was upregulated in the IR zone of control hearts (*p* = 0.005) and was attenuated by melatonin (*p* = 0.350). There were no significant differences in RyR, RyR-*p*, and NCX between the IR and non-IR zones in control and failing hearts. Figure 8B summarizes the results of Na^+^ channel and Cx43 protein expression analyses. There were no significant differences in Cx43, Cx43-p, or SCN5a protein expression between the IR and non-IR zones in any of the four groups. However, the Cx43-p protein level in the control+melatonin group was significantly higher than that in the control group (*p* = 0.047), and the SCN5a protein level was significantly higher in the HF+melatonin group than that in the HF group (*p* = 0.031).

## 3. Discussion

This study investigated the antiarrhythmic mechanisms and molecular targets of 4-week melatonin pretreatment in isolated rabbit hearts with acute IR injury undergoing TH. The main findings are: (1) In failing hearts, melatonin suppressed SDA induction mainly by modulating Ca_i_ dynamics to accelerate Ca_i_ decay, thereby significantly reducing VF maintenance. The molecular targets included the upregulation of SERCA2a and CASQ2 protein expression and the attenuation of the downregulation of PLN-s and PLN-t protein expression in the IR zone. Melatonin also significantly increased SCN5a protein level, but CV did not increase accordingly. (2) In control hearts, melatonin significantly reduced VF maintenance by ameliorating conduction, slowing and lengthening APD_min_ to reduce APD_dispersion_. The molecular targets included increasing the Cx43-p protein level, attenuating the downregulation of SERCA2a and PLN-s protein expression, and attenuating the upregulation of CaMKII-p in the IR zone. (3) TH significantly retarded Ca_i_ decay, decelerated CV, and increased APD_dispersion_, thereby facilitating SDA induction, which counteracted the beneficial effects of melatonin in reducing VF maintenance.

### 3.1. Electrophysiological Targets of Melatonin in Acute IR-Injured Hearts

The impaired function of SERCA2a to sequester cytosolic Ca^2+^ results in Ca_i_ overload and is associated with IR-induced VA [21,22]. Besides its wide spectrum scavenging ability, melatonin was reported to promote SERCA2a expression via the ERK1 pathway to reduce Ca_i_ overload in cardiomyocytes against hypoxia-reoxygenation [23]. Consistently, our data show that 4-week melatonin treatment not only upregulated SERCA2a but also upregulated CASQ2 and attenuated the downregulation of PLN-s and PLN-t protein expression in the IR zone in failing rabbit hearts. CASQ2, the main Ca^2+^-binding protein of the SR, serves as an important regulator of Ca^2+^ to prevent spontaneous SR Ca^2+^ release and triggered arrhythmias [24]. Phosphorylation of PLN releases its inhibitory effects on SERCA2a. The improved SR Ca^2+^ release and reuptake could suppress SDA induction via Ca^2+^-APD coupling [25]. As shown in Figure 1 and Figure 4, melatonin treatment significantly accelerated Ca_i_ decay and suppressed SDA induction, which at least partly underlies the antiarrhythmic mechanism of melatonin in failing hearts with acute IR injury.

CV is primarily regulated by *I*_Na_ excitability, gap junction intercellular conductance, membrane resistance, and factors related to source-sink mismatch, such as fiber structure and wavefront geometry. Cx43 dephosphorylation has been reported to contribute to IR arrhythmia and apoptosis in rat hearts [26]. Benova et al. [27] reported that 5-week melatonin administration (40 μg/day during the night in drinking water) attenuated abnormal myocardial Cx43 distribution and upregulated Cx43 mRNA, total Cx43 protein, and its functional phosphorylated forms in spontaneously hypertensive rats. Melatonin activates phospholipase C [28], which may be involved in upregulating protein kinase C to phosphorylate Cx43 [29]. Consistently, our data showed that 4-week melatonin treatment increased the Cx43-p protein level in control IR-injured hearts, which may play a role in ameliorating CV slowing, especially in the IR zone. In addition, cellular coupling can electronically modulate APD and thus dynamically affect the local dispersion of refractoriness. In a computer modeling study, Lesh et al. showed that the higher the coupling resistance between cells, the less the electrotonic interaction and the greater the unmasking of local differences in intrinsic APD [30]. We previously reported that modulation of gap junction conductance with rotigaptide reduced APD_dispersion_ and improved slowed conduction to defer the onset of arrhythmogenic SDA by dynamic pacing and elevate the pacing threshold of VF [31]. The increased Cx43-p protein level may also play an important role in reducing APD_dispersion_ by melatonin treatment in control IR-injured hearts.

The production of oxygen-derived free radicals during reperfusion also inhibits *I*_Na_, resulting in conduction slowing and unidirectional conduction block [6]. Durkina et al. [32] reported that a 1-week melatonin treatment (10 mg/kg/day orally) increased sodium channel protein expression and peak density of *I*_Na_ to increase CV in normal male Wistar rat ventricles. Our data showed that melatonin increased SCN5a protein levels in failing IR-injured hearts; however, CV did not increase accordingly. This discrepancy could be due to the different animal models used in the two studies (normal rats vs. IR-injured HF rabbits) because sodium channel function is impaired under both HF and IR conditions [33], and CV is also influenced by the disruption of tissue architecture in IR-injured hearts.

### 3.2. Controversial Role of Melatonin in Acute IR-Injured Hearts in Species Other than Rats

Most studies showing the beneficial cardioprotective effects of melatonin have used rats as experimental models; however, these effects appear to be inconsistent in other species. Halladin et al. [34] reported that melatonin failed to protect the heart in a closed-chest porcine model of acute myocardial infarction. However, Bernikova et al. [35] reported that melatonin (4 mg/kg intravenously) given at the first minute of ischemia prevented the early ischemic VF by mitigating excessive QRS prolongation in an open-chest porcine model of acute myocardial infarction. In rabbit models, Dave et al. [36] reported that melatonin (10 mg/kg intravenously) administered 10 min before coronary artery ligation (for 30 min) or 15 min before reperfusion (for 3 h) had no significant effects on the hemodynamic parameters or infarct size. However, Bernikova et al. [37] reported that melatonin (10 mg/kg intravenously) administered 60 min before coronary artery ligation ameliorated APD shortening during ischemia, reduced dispersion of repolarization, and accelerated epicardial pulse conduction in rabbit hearts undergoing 15-min coronary artery ligation and 15-min reperfusion. Our data showed that 4-week melatonin pretreatment (10 mg/kg/day orally) significantly ameliorated conduction slowing and reduced APD_dispersion_ by increasing Cx43-p in control hearts [38]. Furthermore, melatonin improved Ca_i_ dynamics in failing hearts. Different experimental protocols (duration of ischemia, reperfusion, timing and dosing of melatonin administration, regional or global ischemia, and animal species) resulted in different outcomes in IR-injured hearts [39]. The different molecular targets of melatonin’s antiarrhythmic effects in failing and controlling hearts shown in this study warrant further investigation.

### 3.3. Cardioprotective Role of Melatonin in Acute Myocardial IR Injury in Humans

IR injury is one of the most significant sequels of coronary artery bypass graft (CABG) surgery. A meta-analysis study revealed that melatonin supplements could be helpful in reducing cardiac injury and inflammatory biomarkers in patients undergoing CABG [40]. Note that the daily melatonin consumption in clinical studies ranged from 3 to 20 mg/day, which was much lower than the doses used in animal studies, and the observed effects in these clinical studies were also small. Recently, a prospective randomized, single-blinded, placebo-controlled study was conducted to investigate the effects of high-dose melatonin pretreatment (60 mg/day for 5 days) in patients undergoing CABG and its effects on clinical outcomes and markers of apoptosis and inflammation [41]. The results of the study showed that high-dose melatonin reduced serum levels of nuclear factor-κB, tumor necrosis factor-α, interleukin-6, and troponin-I, shortened intubation time, and improved the quality of recovery compared to placebo, with no reported side effects. That is, high-dose melatonin was well tolerated and could ameliorate myocardial IR injury via its antiapoptotic and anti-inflammatory effects to improve clinical outcomes. Further larger-scale clinical studies exploring the dose-dependent effects of melatonin are needed to assess its protective effects and provide stronger evidence for reducing myocardial IR injury.

### 3.4. TH Influences the Antiarrhythmic Effects of Melatonin in Acute IR-Injured Hearts

Our data showed that TH reduced VF maintenance in control IR-injured hearts but had no significant effects on VF maintenance in failed IR-injured hearts. Although TH can induce cell-to-cell uncoupling and Ca_i_ overload to slow conduction, enhance APD_dispersion_, and promote SDA [19,20], TH also exerts its antiarrhythmic effects by prolonging APD and ERP to reduce the vulnerability to reentry, and by slowing RyR release kinetics to prevent triggered activity [42]. As the persistence of fibrillation depends on a critical amount of tissue [43], the prolonged APD induced by TH was more likely to stop VF in smaller control hearts than in larger failing hearts. However, TH counteracted the mechanisms by which melatonin reduced VF maintenance. For example, hypothermia retards SERCA2a-mediated Ca^2+^ uptake [20], thereby counterbalancing the beneficial effects of melatonin on modulating Ca_i_ dynamics in failing IR-injured hearts; hypothermia-induced Cx43 gap junction remodeling [44] also antagonized melatonin’s antiarrhythmic effects in control IR-injured hearts. Because the conductance of gap junctional membranes is temperature dependent [45], the melatonin-increased Cx43-p protein could not function properly at TH, thereby counteracting the beneficial effect of melatonin in reducing APD_dispersion_ in control IR-injured hearts. Although melatonin combined with TH may synergistically exert anti-inflammatory and antiapoptotic effects to reduce IR injury, the combined treatment appears to have no benefit in preventing IR-induced arrhythmias.

## 4. Materials and Methods

The research protocol was approved by the Institutional Animal Care and Use Committee of Chang Gung Memorial Hospital, Taiwan (approval No. 2022083001) and conformed to the Guide for the Care and Use of Laboratory Animals published by the United States National Institutes of Health. Experiments were performed on 40 New Zealand white rabbits (body weight 2.9 to 4.1 kg), randomly divided into HF, HF+melatonin, control, and control+melatonin groups (*n* = 10 in each group). In the HF+melatonin group, melatonin treatment (10 mg/kg/day orally for 4 weeks) [46] began on the same day as pacing was initiated.

### 4.1. Pacing-Induced Heart Failure

Rapid right ventricular pacing was used to induce HF, as described previously [47]. Briefly, the rabbits were premedicated with intramuscular injections of zoletil (15 mg/kg) and xylazine (5 mg/kg). The surgical procedure was performed under general anesthesia using isoflurane (2%). An epicardial pacing lead was fixed on the right ventricle epicardium through a right lateral thoracotomy. The lead was connected to a modified pacemaker (Adapta, Medtronic, Minneapolis, MN, USA). After a one-week recovery period, the heart was paced at a fixed rate of 320 bpm for 4 weeks to induce HF. Left ventricular (LV) function was assessed using echocardiography before and 4 weeks after pacing.

### 4.2. In-Vivo IR Model Creation

Rabbits were premedicated with intramuscular injections of zoletil (15 mg/kg) and xylazine (5 mg/kg), intubated, and anesthetized using isoflurane (2%). The chest was opened through left thoracotomy, and an obtuse marginal branch of the left circumflex artery was ligated midway between the atrioventricular groove and apex for 60 min, followed by reperfusion. If spontaneous ventricular fibrillation (VF) was induced, external defibrillation at 10–25 J was performed.

### 4.3. Langendorff Heart Preparation and Optical Mapping

After a 30-min reperfusion period, the hearts were excised and Langendorff-perfused with 37 °C Tyrode’s solution (composition in mmol/L: NaCl 125, KCl 4.5, MgCl_2_ 0.25, NaHCO_3_ 24, NaH_2_PO_4_ 1.8, CaCl_2_ 1.8, glucose 5.5, and albumin 50 mg/L in deionized water) and equilibrated with 95% O_2_ and 5% CO_2_ to maintain a pH of 7.4. Rhod-2AM (Ca_i_ indicator, 5 μM, Molecular Probes, OR, USA; in 20% pluronic F-127 dissolved in dimethyl sulfoxide) and RH237 (membrane voltage [V_m_] indicator, 1 μM, Molecular Probes; dissolved in dimethyl sulfoxide) were administered. Coronary perfusion pressure was regulated and maintained at 80–90 cmH_2_O. The hearts were illuminated with a solid-state, frequency-doubled laser light source (Millennia, Spectra-Physics Inc., Newport Corporation, Irvine, CA, USA) at a wavelength of 532 nm. Epifluorescence was acquired and filtered (715 mm for V_m_ and 580 nm for Ca_i_) with two MiCAM Ultima cameras (BrainVision, Tokyo, Japan) at 2 ms/frame temporal resolution and 100 × 100 pixels with a spatial resolution of 0.30 × 0.30 mm^2^ per pixel. Motion artifacts were suppressed using blebbistatin (10 μmol/L; Tocris Bioscience, Minneapolis, MN, USA).

### 4.4. Experimental Protocols

A bipolar catheter was inserted into the right ventricle for pacing at twice the threshold. The ventricular effective refractory period (ERP) was measured by giving a premature stimulus after 8 beats at a pacing cycle length (PCL) = 400 ms. Action potential duration (APD) and Ca_i_ alternans were induced by dynamic pacing [47]. VF was induced by an extra stimulus pacing protocol (up to S_5_). Quantification of VF maintenance was based on the VF duration, which was classified as score 0: <10 s, 1: 10–30 s, and 2: ≥30 s, with larger values indicating greater severity [48]. Defibrillation using epicardial patch electrodes was performed if VF persisted > 120 s.

Two thermostatic systems were connected in parallel to the Langendorff system [31]. Induction of TH was performed by switching the thermostatic system and replacing the superfusate to 33 °C. During cooling, LV temperature was monitored continuously with a thermometer implanted into the LV chamber via left atriotomy. Electrophysiological studies were repeated after 30 min of hypothermia.

### 4.5. Western Blot Examination

Cardiac tissues were sampled from the non-IR and IR zones at the end of the mapping studies as previously described (*n* = 6 per group) [47]. Briefly, 50 μg of protein was loaded into each well in the SDS-PAGE gel for electrophoresis. The proteins were then transferred to polyvinylidene difluoride membranes (Immobilon-P; EMD Millipore, Temecula, CA, USA). Primary antibodies against SERCA2a (Thermo, Waltham, MA, USA), phospholamban (PLN, Badrilla, Leeds, UK), pSer16-PLN (PLN-s, Badrilla), pThr17-PLN (PLN-t; Badrilla), calsequestrin 2 (CASQ2, Gene Tex, Irvine, CA, USA), Ca^2+^-calmodulin-dependent protein kinase II (CaMKII, Cell Signaling, Danvers, MA, USA), pThr286-CaMKII (CaMKII-p, Thermo), NCX (Abcam, UK), RyR (ABclonal, Woburn, MA, USA), pSer2808-RyR (RyR-p, LSBio, Seattle, WA, USA), total connexin43 (Cx43, Abcam, Waltham, MA, USA), phosphorylated Cx43 (Cx43-p, Gene Tex), Nav1.5 (SCN5a, Novusbio, Centennial, CO, USA), and β-actin (Gene Tex) were used to detect the proteins of interest. Secondary antibodies, such as goat anti-mouse IgG-horseradish peroxidase (Leinco Technologies, Fenton, MO, USA), goat anti-rabbit IgG-horseradish peroxidase (Leinco Technologies), and donkey anti-goat IgG-horseradish peroxidase antibody (Santa Cruz Biotechnology, Dallas, Texas, USA), were used in conjunction with primary antibodies. Signals were obtained using enhanced chemiluminescence (Pierce ECL Western Blotting Substrate, Thermo Fisher Scientific, Waltham, MA, USA), and the blots were quantified by scanning densitometry. Protein expression levels were normalized to that of β-actin.

### 4.6. Data Analysis

APD was measured at the level of 80% of repolarization. APD_dispersion_ was defined as the difference between maximal (APD_max_) and minimal APD (APD_min_) from the entire mapped areas at PCL = 300 ms [49]. Monoexponential fittings were used to compute the time constant tau (τ) value of the decay portion of the Ca_i_ transient between 70% of the transient peak and the diastolic baseline. The thresholds of spatially concordant alternans (SCA) of APD and Ca_i_ were defined as the longest PCL required to produce a10-ms difference in APD and a 10% difference in Ca_i_ amplitude between consecutive beats, respectively. The phase was considered positive for a short–long APD and a small–large Ca_i_ amplitude sequence (colored in red) and negative for a long–short APD and a large–small Ca_i_ amplitude sequence (colored in green). SDA was evidenced by the presence of both red and green regions separated by a nodal line. The SDA threshold was defined as the longest PCL required to reach the alternans threshold on both sides of a nodal line. To estimate conduction velocity (CV), the distance and conduction time between the earliest activation point and two epicardial points in the non-IR (CV_non-IR_) and IR (CV_IR_) zones were measured [47].

### 4.7. Statistical Methods

Continuous variables are expressed as mean ± standard deviation, and categorical variables are presented as numbers and percentages. Student’s *t*-test was performed to analyze differences in the effective refractory period (ERP), APD, CV, and Ca_i_ decay between the melatonin treatment and non-melatonin treatment groups and between normothermia and TH, and to compare CV and Ca_i_ decay between the non-IR and IR zones. Categorical variables were evaluated using Fisher’s exact test. Statistical analyses were performed using IBM SPSS V25.0 (Armonk, NY, USA). Differences were considered statistically significant at *p* < 0.05.

## 5. Conclusions

Melatonin improves Ca_i_ homeostasis in failing rabbit hearts and ameliorates conduction slowing and APD_dispersion_ in control rabbit hearts to prevent IR-induced VF. These beneficial effects are counteracted by TH.

## Figures and Tables

**Figure 1 ijms-26-00615-f001:**
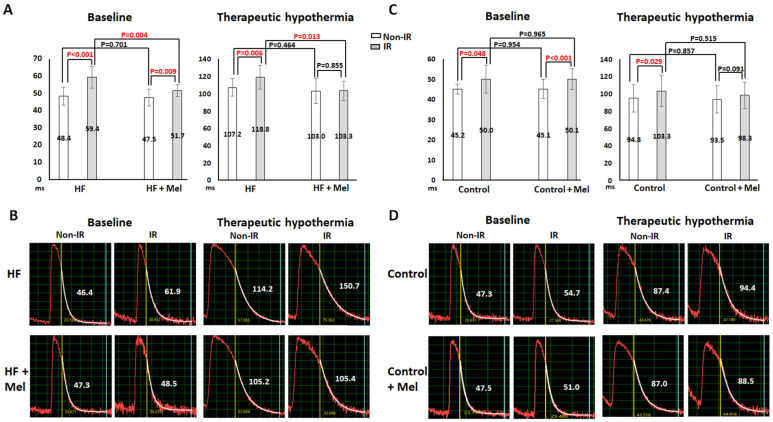
Effects of melatonin (Mel) and therapeutic hypothermia (TH) on intracellular Ca^2+^ (Ca_i_) decay. (**A**,**C**) Summarized results of Ca_i_ decay tau (τ) values in the ischemia–reperfusion (IR) and non-IR zones in heart failure (HF) and control groups, respectively. (**B**,**D**) Representative examples of Ca_i_ decay in HF and control groups, respectively. Numbers indicate the mean values (in ms). TH significantly increased the tau values in the non-IR and IR zones. Melatonin significantly fastened Ca_i_ decay in the IR zone at normothermia and TH in failing hearts.

**Figure 2 ijms-26-00615-f002:**
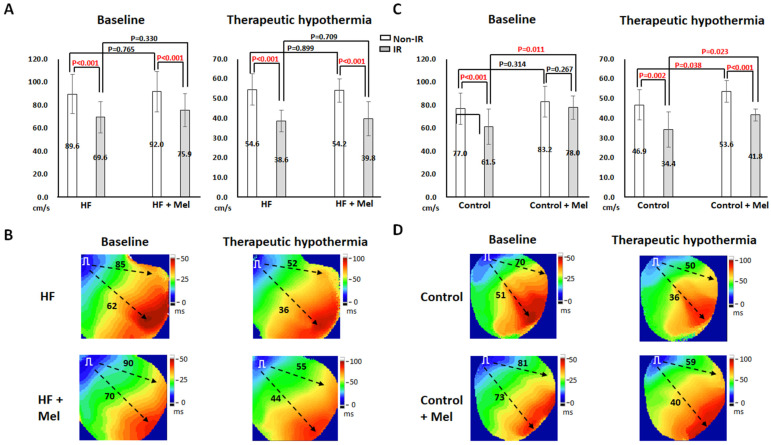
Effects of melatonin (Mel) and therapeutic hypothermia (TH) on conduction velocity (CV). (**A**,**C**) Summarized results of CV in the ischemia–reperfusion (IR) and non-IR zones in heart failure (HF) and control groups, respectively. (**B**,**D**) Representative examples of isochronal maps in HF and control groups, respectively. Numbers indicate the mean values of CV (in cm/s). Dashed black arrows indicate the directions of wavefront propagation.

**Figure 3 ijms-26-00615-f003:**
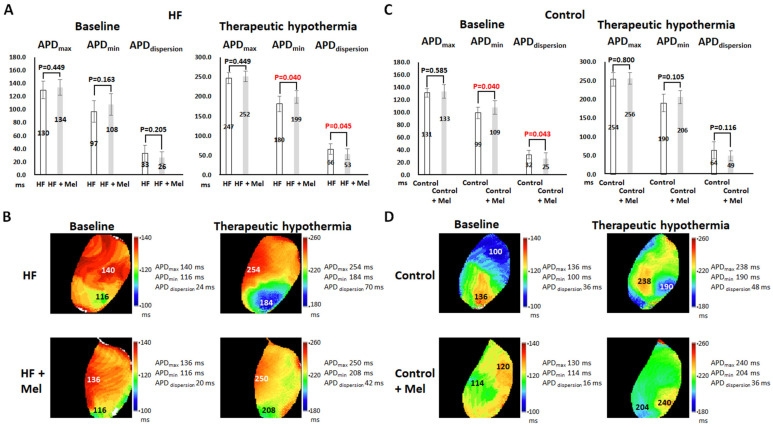
Effects of melatonin (Mel) on action potential duration (APD) at baseline and therapeutic hypothermia (TH). (**A**,**C**) Summarized results of maximal APD (APD_max_), minimal APD (APD_min_), and APD dispersion (APD_dispersion_) at baseline and TH with or without melatonin treatment in heart failure (HF) and control groups, respectively. (**B**,**D**) Representative APD maps in HF and control groups, respectively. Melatonin prolonged APD_min_ and thereby reduced APD_dispersion_ in failing hearts at TH and in control hearts at normothermia. Numbers indicate the values of APD at sites of APD_max_ and APD_min_ in each map (in ms).

**Figure 4 ijms-26-00615-f004:**
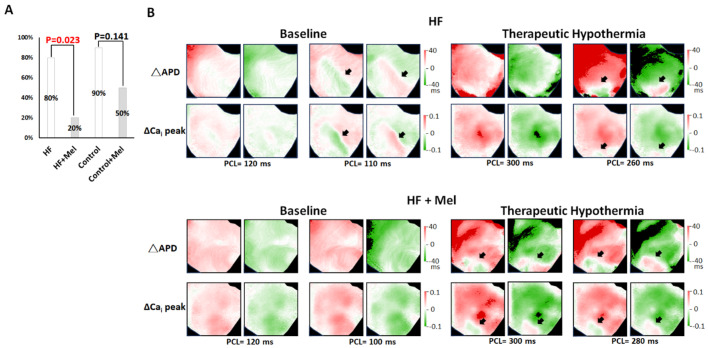
Effects of melatonin (Mel) and therapeutic hypothermia (TH) on spatially discordant alternans (SDA) induction. (**A**) Summarized results of SDA inducibility. (**B**) Representative examples of action potential duration (APD) and intracellular Ca^2+^ (Ca_i_) alternans maps in heart failure (HF) (**upper**) and HF+Mel (**bottom**) groups. Black arrows indicate nodal lines. ΔAPD and ΔCa_i_ are the differences between the two consecutive APD and Ca_i_ amplitudes, respectively.

**Figure 5 ijms-26-00615-f005:**
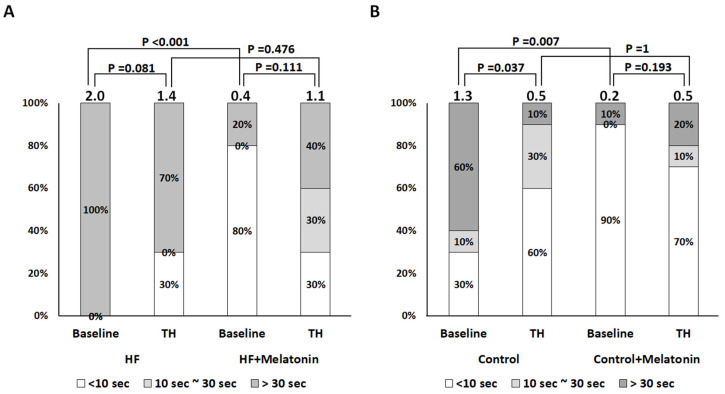
Effects of melatonin and therapeutic hypothermia (TH) on ventricular fibrillation (VF) severity. (**A**,**B**) Summarized results of VF severity in heart failure (HF) and control groups, respectively. Dark grey, light grey, and white color represent VF severity scores 2, 1, and 0, respectively, and the number at the top of each bar represents the mean of VF severity score. Melatonin significantly reduced VF severity in HF and control groups at baseline, which was counteracted by TH.

**Figure 6 ijms-26-00615-f006:**
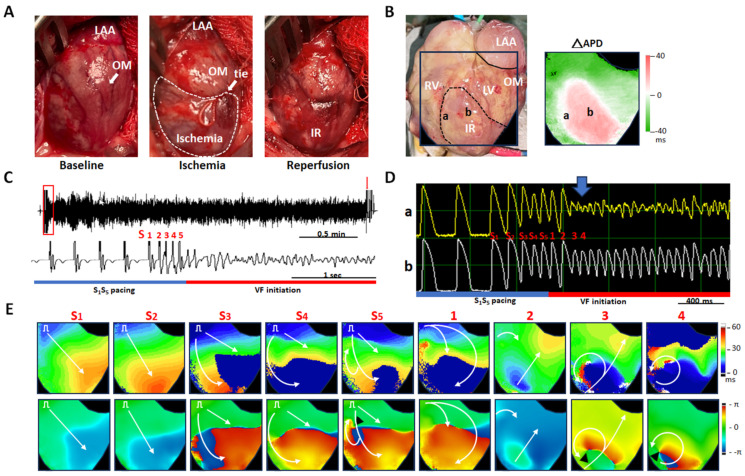
Ventricular fibrillation (VF) induction in a failing heart with acute ischemia-reperfusion (IR)-injury. (**A**) Photos of IR induction through obtuse marginal (OM) branch ligation and release. (**B**) Mapping field (**left**) and action potential duration alternans map (ΔAPD, **right**). (**C**) Pseudo-electrocardiograms show VF induction by extra stimulus pacing. The red arrow indicates a shock spike. The bottom pseudo-electrocardiogram corresponded to the period labeled by a red square in the upper pseudo-electrocardiogram. (**D**) Membrane voltage tracings during VF induction at sites “a” and “b” labeled in Panel (**B**). (**E**) Isochronal maps (**upper**) and phase maps (**bottom**) of VF induction (labeled in Panel (**D**)). White arrows indicate multiple impulses within the mapping field; black triangles indicate phase singularities. LAA, left atrial appendage; LV, left ventricle; RV, right ventricle.

**Figure 7 ijms-26-00615-f007:**
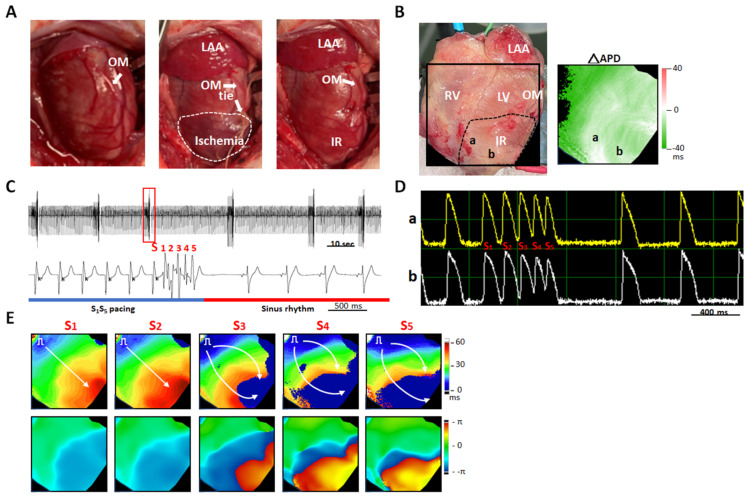
Melatonin pretreatment suppressed ventricular fibrillation induction in a failing heart with acute ischemia-reperfusion (IR) injury. (**A**) Photos of IR induction through obtuse marginal (OM) branch ligation and release. (**B**) Mapping field (**left**) and action potential duration alternans map (ΔAPD, **right**). (**C**) Pseudo-electrocardiograms show that extra stimulus pacing failed to induce VF. The bottom pseudo-electrocardiogram corresponded to the period labeled by a red square in the upper pseudo-electrocardiogram. (**D**) Membrane voltage tracings during extrastimulus pacing at sites “a” and “b” labeled in Panel (**B**). (**E**) Isochronal maps (**upper**) and phase maps (**bottom**) of extra stimulus pacing (labeled in Panel (**D**)). White arrows indicate impulse propagation within the mapping field. LAA, left atrial appendage; LV, left ventricle; RV, right ventricle.

**Figure 8 ijms-26-00615-f008:**
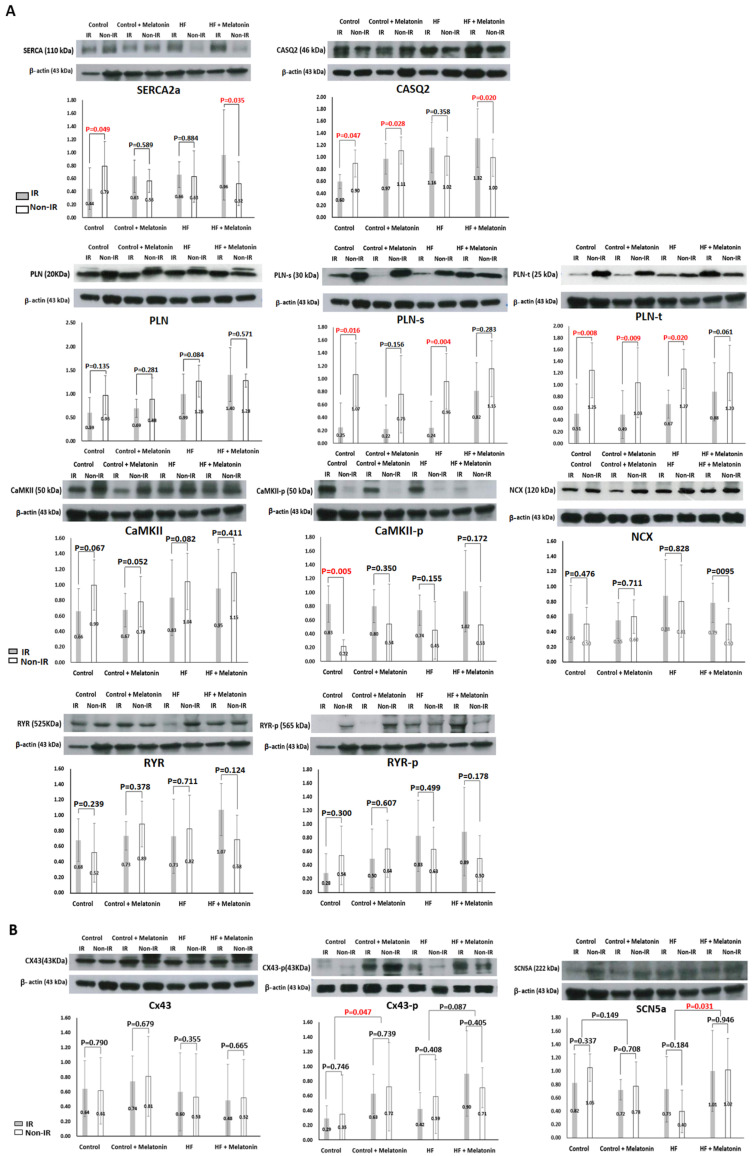
Results of protein analyses. Representative bands (upper subpanels) and summarized results of densitometric values normalized to the corresponding β-actin (lower subpanels). CaMKII, Ca^2+^-calmodulin-dependent protein kinase II; CaMKII-p, pThr286-CaMKII; CASQ2, calsequestrin 2; Cx43, connexin 43; Cx43-p, phosphorylated Cx43; HF, heart failure; IR, ischemia-reperfusion; NCX, sodium-calcium exchanger; PLN, phospholamban; PLN-s, pSer16-PLN; PLN-t, pThr17-PLN; RyR, ryanodine receptor 2; RyR-p, phosphorylated RyR2; SERCA2a, sarcoplasmic reticulum Ca^2+^-ATPase.

**Table 1 ijms-26-00615-t001:** Electrophysiological effects of melatonin and therapeutic hypothermia in isolated rabbit hearts with acute ischemia-reperfusion injury.

		APD_max_ (ms)	APD_min_ (ms)	APD_diff_ (ms)	ERP (ms)	SCA Threshold (ms)	SDA Threshold (ms)		Ca_i_ Decay (ms)	CV_300_ (cm/s)
Baseline	HF (*n* = 10)	130 ± 14	97 ± 16	33 ± 12	135 ± 18	145 ± 21	113 ± 13 (*n* = 8)	Non-IR	48.4 ± 5.2	89.6 ± 17.0
								IR	59.4 ± 6.4 ‡	69.6 ± 13.6 ‡
	HF + Mel (*n* = 10)	134 ± 12	108 ± 17	26 ± 9	135 ± 16	137 ± 17	105 ± 7 (*n* = 2)	Non-IR	47.5 ± 4.9	92.0 ± 17.5
								IR	51.7 ± 3.5 *‡	75.9 ± 13.6 ‡
Therapeutic hypothermia	HF (*n* = 10)	247 ± 15 †	180 ± 20 †	66 ± 13 †	285 ± 22 †	390 ± 32 †	258 ± 34 (*n* = 10)	Non-IR	107.2 ± 10 †	54.6 ± 7.8 †
								IR	118.8 ± 13.8 †‡	38.6 ± 5.4 †‡
	HF + Mel (*n* = 10)	252 ± 14 †	199 ± 17 *†	53 ± 14 *†	290 ± 35 †	389 ± 33 †	264 ± 37 (*n* = 10)	Non-IR	103.0 ± 14.7 †	54.2± 5.9 †
								IR	103.3 ± 11.3 *†	39.8 ± 8.6 †‡
Baseline	Control (*n* = 10)	131 ± 7	99 ± 9	32 ± 7	123 ± 14	142 ± 19	113 ± 13 (*n* = 9)	Non-IR	45.2 ± 2.5	77.0 ± 13.4
								IR	50.0 ± 6.9 ‡	61.5 ± 15.3 ‡
	Control + Mel (*n* = 10)	133 ± 11	109 ± 10 *	25 ± 9 *	129 ± 13	139 ± 14	118 ± 16 (*n* = 4)	Non-IR	45.1 ±4.9	83.2 ± 13.4
								IR	50.1 ± 5.2 ‡	78.0 ± 10.2 *
Therapeutic hypothermia	Control (*n* = 10)	254 ± 19 †	190 ± 24 †	64 ± 24 †	260 ± 16 †	381 ± 34 †	256 ± 41 (*n* = 10)	Non-IR	94.8 ± 15.7 †	46.9 ± 7.7 †
								IR	103.3 ± 18.2 †‡	34.4 ± 8.9 †‡
	Control + Mel (*n* = 10)	256 ± 16 †	206 ±17 †‡	49 ± 13 †‡	258 ± 15 †	395 ± 16 †	263 ± 50 (*n* = 10)	Non-IR	93.5 ± 15.8 †	53.6 ± 5.6 *†
								IR	98.3 ± 15.3 †	41.8 ± 3.1 *†‡

Values are mean ± standard deviation. APD_max_, APD_min_ and APD_diff_, maximal, minimal and difference of action potential duration at pacing cycle length of 300; CV_300_, conduction velocity at pacing cycle length of 300; Ca_i_, intracellular Ca^2+^; ERP, effective refractory period; HF, heart failure; IR, ischemia-reperfusion; SCA, spatially concordant alternans; SDA, spatially discordant alternans, TH, therapeutic hypothermia. * *p* < 0.05, non-melatonin vs. melatonin; † *p* < 0.05, baseline vs. TH; ‡ *p* < 0.05, non-IR vs. IR.

## Data Availability

The data that support the findings of this study are included in the article and Appendix A.

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
