# Peer review of "Mechanistic Insights into Melatonin’s Antiarrhythmic Effects in Acute Ischemia-Reperfusion-Injured Rabbit Hearts Undergoing Therapeutic Hypothermia"

_ijms, 2025, doi:10.3390/ijms26020615_

Round 1

Reviewer 1 Report

Comments and Suggestions for Authors

I read with great interest this manuscript on melatonin to reduce arrhythmias induced by ischemia-reperfusion injury.

The manuscript is well-written and I only have 2 conceptual remarks/questions:

1. The dose of melatonin seems rather high. How does it compare to doses used in humans?

2. VF severity is a rare term. There is no magnitude of VF only inducibility or duration could be different even though VF rarely terminates spontaneously in humans.

Author Response

I read with great interest this manuscript on melatonin to reduce arrhythmias induced by ischemia-reperfusion injury.

The manuscript is well-written and I only have 2 conceptual remarks/questions:

  1. The dose of melatonin seems rather high. How does it compare to doses used in humans?
    Reply: Thanks for the reviewer’s comment. We followed the protocol of a previous paper investigating melatonin in a high-fat diet rabbit model (new reference #46) and administered melatonin orally at a dose of 10 mg/kg/day. In clinical studies, the daily melatonin consumption doses ranged from 3 to 20 mg/day, (new reference #40) that is, the estimated doses in humans were around 0.05 to 0.3 mg/kg/day. Recently, a prospective randomized, single-blinded, placebo-controlled study was conducted to investigate the effects of high-dose melatonin pretreatment (60 mg/day for 5 days) in patients undergoing CABG and its effects on clinical outcomes and markers of apoptosis and inflammation. (new reference #41) The results of the study showed that high-dose melatonin reduced serum levels of nuclear factor-κB, tumor necrosis factor-α, interleukin-6, and troponin-I, shortened intubation time, and improved the quality of recovery compared to placebo, with no reported side effects. The estimated dose was around 1 mg/kg/day. We do not know whether oral melatonin administration at 1 mg/kg/day is effective in preventing IR-induced VF in our rabbit models. Further larger-scale clinical studies exploring the dose-dependent effects of melatonin are needed to access its protective effects and provide stronger evidence for reducing myocardial IR injury (page 23, line 13 to page 24 line 11).  
  1. VF severity is a rare term. There is no magnitude of VF only inducibility or duration could be different even though VF rarely terminates spontaneously in humans.
    Reply: Thanks for the reviewer’s comment. We change “VF severity” to “VF maintenance”. 

Reviewer 2 Report

Comments and Suggestions for Authors

This study offers a detailed examination of melatonin's antiarrhythmic effects on ischemia-reperfusion injury in rabbit hearts, carefully documenting cellular mechanisms and electrophysiological findings. The inclusion of therapeutic hypothermia adds further depth and relevance to the investigation. Nonetheless, a few aspects could benefit from additional consideration:

1. The discussion on calcium handling proteins and gap junctions is insightful, but stronger connections between these molecular changes and specific electrophysiological observations, such as action potential variations, would be valuable.

2. While the use of rabbit hearts provides useful insights, discussing the broader applicability of these findings to other models or clinical contexts would strengthen the manuscript. Referencing studies involving human subjects or different species could offer a more comprehensive perspective.

3. The observation of hypothermia diminishing melatonin's protective effects is notable. Expanding on the molecular mechanisms driving this interaction, particularly how hypothermia-induced cell uncoupling affects melatonin's action, could clarify these findings further.

4. Some results are described as significant without consistently providing corresponding p-values or effect sizes. Including these metrics throughout the text would improve the clarity and rigor of the analysis.

5. The figures presented are informative, but certain areas could benefit from clearer labeling, especially regarding the regions of interest and the comparisons being highlighted.

Author Response

This study offers a detailed examination of melatonin's antiarrhythmic effects on ischemia-reperfusion injury in rabbit hearts, carefully documenting cellular mechanisms and electrophysiological findings. The inclusion of therapeutic hypothermia adds further depth and relevance to the investigation. Nonetheless, a few aspects could benefit from additional consideration:

  1. The discussion on calcium handling proteins and gap junctions is insightful, but stronger connections between these molecular changes and specific electrophysiological observations, such as action potential variations, would be valuable.
    Reply: Thanks for the reviewer’s comment. We add discussion on gap junction modulation as one of the mechanisms underlying reducing APDdispersion by melatonin in control IR-injured hearts (page 21, lines 11-19).
  1. While the use of rabbit hearts provides useful insights, discussing the broader applicability of these findings to other models or clinical contexts would strengthen the manuscript. Referencing studies involving human subjects or different species could offer a more comprehensive perspective.
    Reply: We add a new paragraph in the Discussion section “Cardioprotective role of melatonin in acute myocardial IR injury in humans” to reference studies involving human subjects (page 23, line 13 to page 24 line 11).
  1. The observation of hypothermia diminishing melatonin's protective effects is notable. Expanding on the molecular mechanisms driving this interaction, particularly how hypothermia-induced cell uncoupling affects melatonin's action, could clarify these findings further.
    Reply: We have explained how therapeutic hypothermia counteracted the mechanisms by which melatonin reduced VF maintenance. For example, hypothermia retards SERCA2a-mediated Ca2+ uptake, thereby counterbalancing the beneficial effects of melatonin on modulating Cai dynamics in failing IR-injured hearts; hypothermia-induced Cx43 gap junction remodeling also antagonized melatonin’s antiarrhythmic effects in control IR-injured hearts (page 25, lines 3-6). To further clarify these findings, we add “Because the conductance of gap junctional membranes is temperature dependent, the melatonin-increased Cx43-p protein could not function properly, thereby counteracting the beneficial effect of melatonin in reducing APDdispersion in control IR-injured hearts.” (page 25, lines 6-9).
  1. Some results are described as significant without consistently providing corresponding p-values or effect sizes. Including these metrics throughout the text would improve the clarity and rigor of the analysis.
    Reply: The animal numbers were 10 and 6 in each group for comparisons of optical mapping data and western blot data, respectively. We have added p values for all comparisons in the Result section.
  1. The figures presented are informative, but certain areas could benefit from clearer labeling, especially regarding the regions of interest and the comparisons being highlighted.
    Reply: We added maximal and minimal APD values in Figures 3B and 3D to label the sites of APDmax and APDmin in each map clearly.